# Dentin Bonding Performance of Universal Adhesives in Primary Teeth In Vitro

**DOI:** 10.3390/ma16175948

**Published:** 2023-08-30

**Authors:** Nina Danevitch, Roland Frankenberger, Susanne Lücker, Ulrich Gärtner, Norbert Krämer

**Affiliations:** 1Department of Pediatric Dentistry, Medical Center for Dentistry, University Medical Center Giessen and Marburg, 35392 Giessen, Germany; 2Department of Operative Dentistry, Endodontics, and Pediatric Dentistry, Medical Center for Dentistry, University Medical Center Giessen and Marburg, 35039 Marburg, Germany; 3Institute for Anatomy and Cell Biology, University of Giessen, Aulweg 123, 35392 Giessen, Germany

**Keywords:** dentin, microtensile bond strength, pre-test failure, primary teeth, universal adhesives

## Abstract

(1) Background: The aim of this in vitro study was to evaluate the micro-tensile bond strength (µ-TBS) of universal adhesives to primary tooth dentin after different storage periods. (2) Methods: Dentin of 100 extracted primary molars was exposed. Dentin surfaces were bonded with six universal adhesives (Adhese^®^Universal [AU], All-Bond Universal^®^ [ABU], G-Premio Bond [GPB], iBond^®^Universal [IBU], Prime&Bond active™ [PBa], and Prime&Bond^®^NT as control [PBN]) and restored with a resin composite build-up (Filtek™ Z250). After 24 h, 6 months, and 12 months of water storage, specimens were cut into sticks, and µ-TBS was measured and analyzed using one-way ANOVA (*p* < 0.05) for normal distributions and the Mann–Whitney U-test (*p* < 0.05) for non-normal distribution. Pretesting failures were recorded as 0 MPa. Fracture modes were analyzed under a fluorescence microscope; interfaces were visualized with SEM/TEM. (3) Results: Compared with the reference group (PBN: 32.5/31.2 MPa after 6/12 months), two adhesives showed a significantly higher bond strength after 6 months (AU: 44.1 MPa, ABU: 40.9 MPa; *p* < 0.05) and one adhesive after 12 months (AU: 42.9 MPa, *p* < 0.05). GPB revealed significantly lower bond strengths in all storage groups (16.9/15.5/10.9 MPa after 24 h/6 months/12 months; *p* < 0.05). AU and IBU did not suffer pre-test-failures [PTF]. (4) Conclusions: After 12 months, PBN, IBU, AU, and GPB showed significantly lower results compared ithw initial µ-TBS, whereas AU revealed the highest µ-TBS and no PTF.

## 1. Introduction

Despite scientifically proven success of caries prevention, tooth decay is still one of the most common global diseases [1]. E.g., in Europe, caries prevalence and especially the number of untreated carious lesions in the target group of children < 6 years ranges between 20 and 90% [2]. Therefore, the demand for effective and durable restorations in deciduous teeth is still surprisingly high [1,2].

Due to their characteristic polymerization shrinkage and hydrophobicity, resin composites require a separate adhesive; therefore, these materials are regularly called bonded resin composites (BRC) [3]. During the past few decades, the development of adhesives has been evolving rapidly; however, this development has not aimed for maximum efficacy all the time [4]. For quite a substantial time, when it came to enamel and dentin bonding, speed was prioritized over performance [4]. Although a certain speed is helpful in pediatric dentistry, performance is always more important [2]. The chronological development of dental adhesives started with multi-step systems over 30 years ago, leading to single-bottle systems, and it progressed from etch-and-rinse over selective-etch to self-etch adhesives representing a huge variety of adhesives with the latter being more favorable for primary teeth [2]. Finally, universal adhesives are characterized by their original claim to be equally effective on pre-etched vs. non-etched dentin [5].

The main goal of adhesives in primary teeth is primarily durable dentin adhesion because dentin areas are larger than in permanent teeth and enamel is often severely worn over time [2]. From the in vitro point of view, this makes appropriate investigations regarding long-term dentin bonding quite interesting.

Universal adhesives are classified according to their acidity and consequently observed demineralization depths (Table 1) [6]. A lower pH is associated with a deeper demineralization; however, this is not automatically associated with a more durable bonding performance [5,6]. Today, 10-methacryloyloxydecyl-dihydrogen-phosphate (10-MDP) is regarded successful as a functional monomer with effective etching and dentin protecting performance [7,8,9,10].

Compared with permanent teeth, primary teeth reveal a different macro- and micromorphology. They are less mineralized and, therefore, more prone to demineralization in shorter periods, they have larger diameters of dentinal tubules and less volume of intertubular and peritubular dentin [5,6,11].

The aim of the present study was to compare different universal adhesives in self-etch mode to the former clinical gold standard PBN regarding micro-tensile bond strength (µ-TBS) to dentin of primary teeth after 24 h, 6 and 12 months of water storage. The null hypothesis was that there would be no difference between PBN and the different adhesives both initially and after storage.

## 2. Materials and Methods

After ethical approval (University of Giessen Ethics Committee, Germany, Code 143/09), 90 freshly extracted primary molars were stored in 0.5% chloramine T for less than four weeks. Teeth were ground flat to expose caries-free dentin, simulating caries excavation with a smear layer (Grinder-Polisher Beta, Buehler, Wooster, OH, USA) having been produced with abrasive paper under continuous water cooling (Met II (Grit 360 (P600) and Grit 600 (P1200), Buehler). Sample size was mainly guided by maximal capacity of the experimental setup, but it was also in line with previous studies [5,6,10].

Specimens were bonded (one restorative operator) with six universal adhesives (Table 2; Adhese^®^Universal [AU], All-Bond Universal^®^ [ABU], G-Premio Bond [GPB], iBond^®^Universal [IBU], Prime&Bond active™ [PBa], and Prime&Bond^®^NT as control [PBN]). Table 2 displays the different application details exactly. A resin composite build-up (Filtek™ Z250/3M) was applied (Figure 1). The initial layer of the resin composite was 0.5 mm thick, followed by 1–2 mm layers, each one light-cured for 40 s (bluephase^®^ G2, Ivoclar Vivadent AG, Schaan, Liechtenstein). After 24 h, 6 months, and 12 months of storage in distilled water which was changed monthly (37 °C, Unity^TM^ Lab Services, Thermo Fisher Scientific, Waltham, MA, USA), the samples were cut into sticks, each with a surface area of 0.4225 mm^2^ (Isomet 5000 Linear Precision Saw, IsoMet^TM^ Diamond Wafering Blades (0.4 mm), Buehler; 3450 rpm, 2.5 mm/min, 75 g). With the resulting 1158 sticks, µ-TBS tests were carried out at a crosshead speed of 1 mm/min (TC-550, Version 3.1.0.127, Syndicad, München, Germany).

After debonding, both sides of the fractured specimens were analyzed under a fluorescence microscope at 40× magnification, where dentin and adhesive/composite exhibited different fluorescence values and were easily detectable (Nikon AZ100, Tokyo, Japan). Additionally, selected specimens were immersed in 4% sodium hypochlorite (diluted from 12% NaOCl, Carl Roth, Karlsruhe, Germany) for 20 min and demineralized in 20% hydrochloric acid (diluted from 37% HCl, Sigma-Aldrich, St. Louis, MO, USA) for 6 h, each step followed by rinsing with distilled water. Afterwards, specimens were dehydrated in ascending concentrations of ethanol (70–80–90% for 20 min, 100% for 1 h) and 1,1,1,3,3,3-hexamethyldisilazane for 10 min followed by drying overnight [12]. The specimens were sputter coated with gold (Sputter Coater, Polaron, SC502, Fisons Instruments, Ipswich, UK) and consequently examined under an scanning electron microscope (SEM Amray Model 1610 Turbo, Amray, Bedford, MA, USA).

For transmission electron microscopy (TEM), selected specimens were demineralized in 10% buffered ethylenediaminetetraacetic acid for 72 h. Then, the specimens were fixed in a mixture of 2.5% glutaraldehyde and 2% paraformaldehyde in 0.1 M sodium cacodylate buffer at pH 7.4 for 12 h in 4 °C followed by rinsing with 0.1 M sodium cacodylate buffer for 2 h in 4 °C. Post-fixation with osmium tetroxide in 0.1 M sodium cacodylate buffer for 1 h was executed and rinsed afterwards with 0.1 M sodium cacodylate buffer for 20 min. For drying purposes, ethanol was used in an ascending concentration (50–70–95–100%, each for 1 h). Afterwards, the specimens were immersed in propylene oxide for 20 min. To embed the specimens, they were immersed in a mixture of propylene oxide and epoxy resin (50%:50%) for 6 h in a rotator (6 rpm) followed by infiltration of epoxy resin under vacuum for 12 h. Finally, the specimens were embedded in fresh epoxy resin. The specimens were dried in an oven for 12 h in 65 °C. The dry specimens were cut in 90 nm slabs and dyed with 2% uranyl acetate for 10 min and 3% lead citrate for 5 min. After drying, the specimens were examined under TEM (Zeiss EM 902, Zeiss, Germany).

Statistical analysis was performed by SPSS^®^ 26 (IBM^®^). Normal Distribution was determined with Kolmogorov–Smirnov test. One-way ANOVA test (*p* < 0.05) for normal distributions and the Mann–Whitney U-test (*p* < 0.05) for non-normally distribution were used. The fracture analysis was performed descriptively (Table 3). Therefore, the different fracture modifications were divided into 3 groups: Pretesting failures (0 MPa), adhesive fractures, and cohesive fractures which contain fractures within the resin composite, dentin, or mixed fractures.

## 3. Results

Results are displayed in Table 3 and Figure 2 and Figure 3. The values of µ-TBS ranged between 0 and 95.0 Mpa (IBU after 24 h). After 24 h GPB and PBA, significantly lower results were seen compared with PBN (GPB: 16.9 Mpa; PBA: 31.7 Mpa; one-way ANOVA test, *p* < 0.00, *p* = 0.035). In the 6 months group, ABU and AU showed significantly higher results than PBN, whereas GPB once again revealed the lowest results (AU: 44.1 Mpa, ABU: 40.1 Mpa, GPB: 15.5 Mpa; one way ANOVA test, *p* < 0.00, *p* = 0.033, *p* < 0.00). Consistently, GPB exhibited after 12 months again significant lower µ-TBS than control group (10.9 Mpa; one-way ANOVA test, *p* < 0.00). AU showed significantly higher results after 12 months (42.9 Mpa; one-way ANOVA test, *p* = 0.07). Within the adhesive groups over the storage periods PBN, IBU and AU revealed significant lower results compared with their initial µ-TBS (one-way ANOVA test, PBN: *p* < 0.00, IBU: *p* < 0.00, AU: *p* = 0.01, *p* < 0.00). GPB and CUB showed significant lower results after 12 months and IBU than after 24 h (one-way ANOVA test, *p* < 0.00).

Fluorescence microscopy again helped to visualize exact failure mode determination [12] (Figure 4, Figure 5 and Figure 6). Compared with light microscopy alone, fluorescence microscopy is much easier for the evaluator’s eye to detect different substrates. AU and IBU did not suffer any pretesting failures over the storage periods. GPB showed the highest number of PTF (5% after 6 months, 12% after 12 months).

SEM analysis exhibited significant resin tag formation with various lengths, and different thicknesses of hybrid layers (Figure 7, Figure 8 and Figure 9). Especially after 12 months, tag length in PBA was approximately 10 µm (Figure 9). IBU showed multiple anastomoses between tags (Figure 8). However, micromorphological appearance under the SEM such as resin tag length or hybrid layer thickness, directly correlated with adhesive performance over time. Moreover, the reason may be that all tested adhesives in the present investigation performed quite well to counteract shrinkage stresses occurring in daily situations.

TEM analysis (Figure 10, Figure 11 and Figure 12) exhibited characteristic findings for primary tooth dentin. For ABU, a homogenous adhesive layer with a hybrid layer of approximately 300 nm thickness (Figure 10) was detected, whereas AU revealed inhomogeneous precipitates in the adhesive and a thicker hybrid layer of ca. 500 nm (Figure 11). The adhesive IBU exposed a gradient from dentin to the covering resin composite which had been increasing after 12 months storage and showed a pattern similar to “piano keys” (Figure 12).

## 4. Discussion

The aim of the present study was to evaluate the µ-TBS of different universal adhesives to primary tooth dentin after water storage periods of 24 h, 6, and 12 months. Main disadvantage of these in vitro studies is that they mainly deal with caries-free dentin; however, caries-affected dentin is difficult to obtain in a standardized manner and almost impossible to mimic correctly. Nevertheless, relevant storage periods have been added, and a huge number of beams was tested and analyzed in order to provide reliable results.

Since the introduction of 10-MDP, several studies comparing adhesives with different functional monomers, such as HEMA, 4-META, and PENTA, have been executed either on adult teeth or primary teeth [12,13,14,15].

Different pretreatment modes such as etch-and-rinse and self-etch were analyzed as well. Multi-step adhesives were compared with all-in-one-adhesives; however, especially for primary teeth, further evidence is necessary [12,13,14,15]. It is important for the pediatric dentist to know that the micromorphology of primary teeth is different from permanent teeth. Primary dentin is characterized by less mineralization, less density, and larger dentinal tubules of up to 10 µm in diameter. Mineralization of inter- and peri-tubular dentin of primary teeth is reported to be considerably lower than in permanent teeth [2,6,16]. This also influences the mineralization grade and, therefore, the removability of dentin smear layers [2,16]. Therefore, several studies identified the self-etch approach being the best choice for deciduous teeth’s dentin bonding, making phosphoric acid etching a questionable clinical measure [17].

For this study, we used six 10-MDP containing adhesives and compared them to the former gold standard in primary teeth (PBN). We defined it as the gold standard because its efficacy, ease, and low technique sensitivity are reflected by an array of positive clinical data. The adhesives under investigation differ in several other aspects such as functional monomers, solvent, and pH. With the introduction of 10-MDP by Kuraray Noritake Dental Inc. (Okayama, Japan) in 1981, a functional monomer with a spacer of 10 methyl groups was established (Figure 4). 10-MDP showed better results regarding µ-TBS than functional monomers with more or less methyl groups [18]. One possible reason could be more stable salts with 10-MDP and calcium [19]. Another reasonable explanation could be the so-called nanolayering [20]. This protecting layer of approximately 4 nm out of 10-MDP and calcium salts may impregnate the collagen fibers against hydrolysis and degradation [21].

Besides exogenous factors endogenous factors like endopeptidases are involved in hydrolysis and degradation of the hybrid layer [22,23]. Endopeptidases are proteolytic peptidase of human tissues. Their main function is the physiological but also pathological remodeling of tissues [24]. In human dentin there are matrix metalloproteinases (MMP) like MMP-2, -3, -8, and -9 represented, as well as cysteine cathepsins (CC) [24,25]. The activation of the endopeptidases runs in acidic milieu [26]. Another important and maybe the most important antagonist of clinical long-term performance in dentin bonding is hydrolysis [27]. Many adhesives contain water as solvents. It was previously discussed whether lower µ-TBS could be associated with osmotic processes from intrinsic water, for example bonded or free water on one hand, and extrinsic water in the application process or water storage on the other hand [28]. During bonding application, thorough evaporation of water is necessary. To facilitate this, acetone or/and ethanol are added as additional solvents. In this study, GPB with a rather hydrophobic solvent revealed the lowest µ-TBS and the most PTF, whereas AU based on ethanol/water showed significant higher bond strengths after 6 and 12 months without any PTF compared with PBN with acetone as a solvent.

Furthermore, it is interesting to have a closer look at the pH values of different adhesives. In the present investigation, the lowest pH value of 1.6–1.8 was associated with the best outcomes compared with PBN. However, compared with their initial µ-TBS, we constantly found significantly lower bond strengths after water storage. When used as mild adhesive (pH 2.3–3.0), AU revealed significantly better results than PBN after 6 and 12 months of water storage. Compared with a previous study dealing with different dentin etching times, the present investigation shows promising and quite stable results using the self-etch technique [29]. This is also due to the fact that any etch-and-rinse approach beyond 7 s is currently considered “overetching” in primary teeth dentin and apparently promotes hydrolysis over time [29]. Within the limits of the study, we can conclude that there we identified adhesives having been more effective and stable for bonding to primary tooth dentin compared to the former gold standard by exhibiting the same low technique sensitivity. Finally, the null hypothesis had to be partially rejected. For final conclusions, prospective randomized clinical studies are still the ultimate instrument.

## 5. Conclusions

Compared with the clinically well-proven gold standard Prime&Bond NT, the universal adhesives under investigation showed different results in primary teeth dentin when used in self-etch mode. Despite some differences in adhesive performance over time, all adhesives under investigation are regarded as clinically promising. Nevertheless, bond degradation over time was observed in all groups.

## Figures and Tables

**Figure 1 materials-16-05948-f001:**
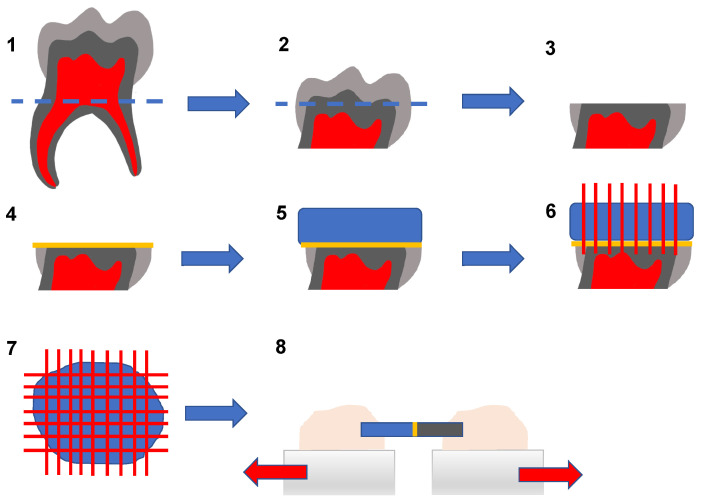
Step 1: Removal of 80% of roots; Step 2/3: grinding flat to expose dentin beneath the fissure; Step 4: adhesive pre-treatment; Step 5: resin composite build-up; Step 6: cutting the core into slabs; Step 7: cutting the resulting slabs into sticks; Step 8: microtensile testing after different storage periods.

**Figure 2 materials-16-05948-f002:**
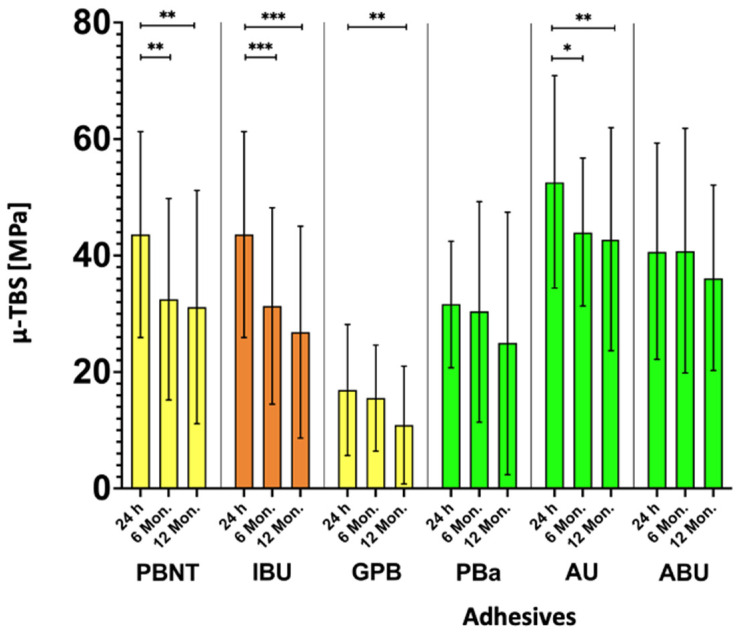
Results for the µ-TBS to dentin compared with PBN. Asterisk indicates a significant difference within the different storage periods (* *p* < 0.05, ** *p* < 0.01, *** *p* < 0.00), sort by pH value (orange = medium, yellow 0 mild, green = ultra-mild).

**Figure 3 materials-16-05948-f003:**
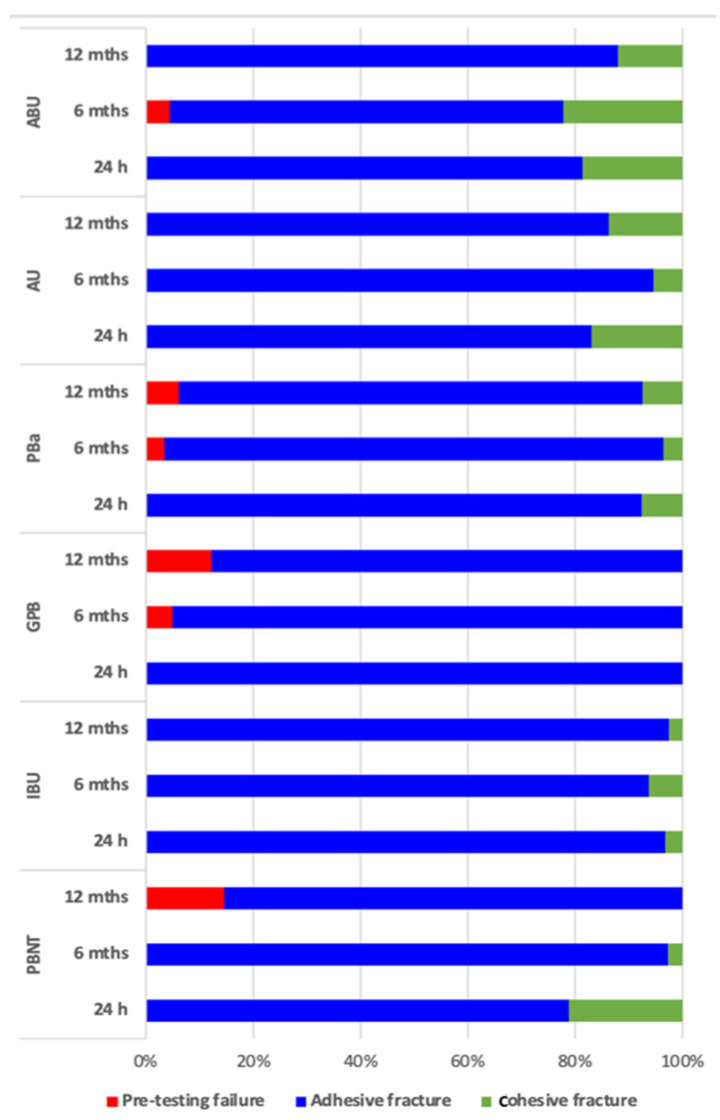
Analysis of fracture failure modifications of the different adhesives investigated regarding the storage periods, sorted by pH value compared with PBN.

**Figure 4 materials-16-05948-f004:**
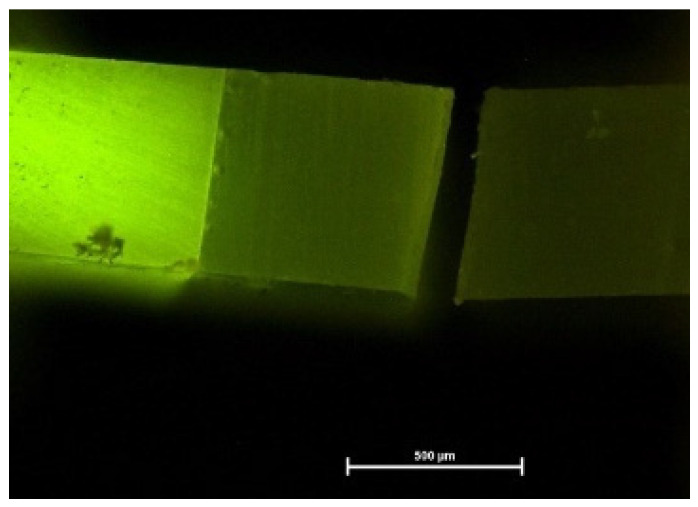
Fluorescence micrograph of a cohesive fracture. PBA after 6 months, fluorescence microscope with 40× magnification (bar: 500 µm).

**Figure 5 materials-16-05948-f005:**
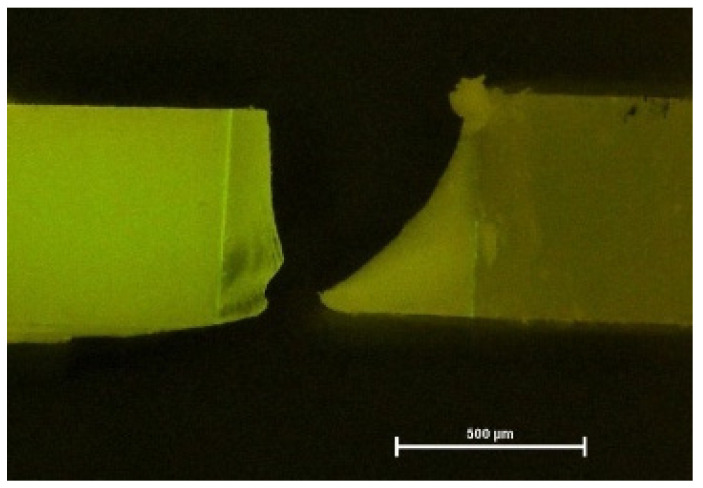
Fluorescence micrograph of a mixed fracture. PBA after 6 months, fluorescence microscope with 40× magnification (bar: 500 µm).

**Figure 6 materials-16-05948-f006:**
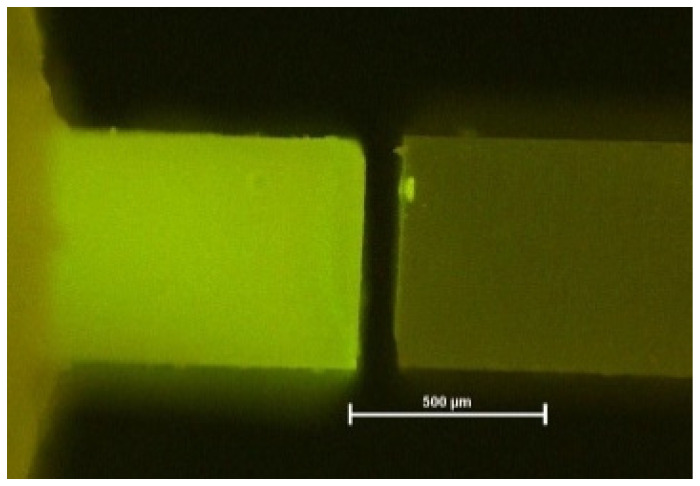
Fluorescence micrograph of an adhesive fracture. PBA after 6 months, fluorescence microscope with 40× magnification (bar: 500 µm).

**Figure 7 materials-16-05948-f007:**
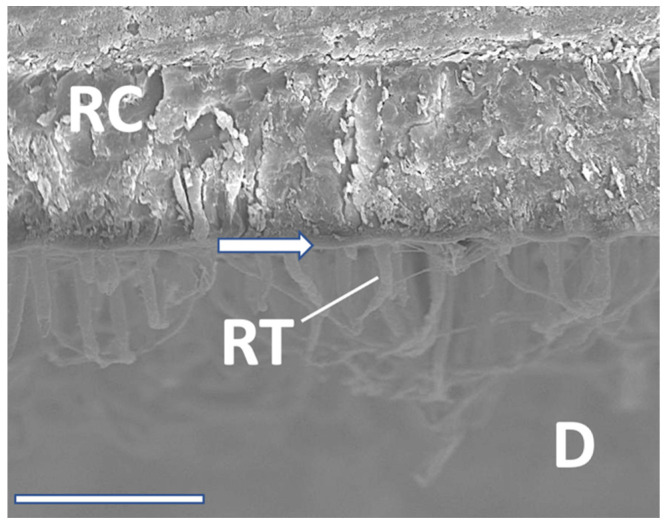
SEM interface (RC: resin composite; RT: resin tags; D: Dentin; hybrid layer: arrow). ABU after 24 h. 1500× magnification/bar: 20 µm.

**Figure 8 materials-16-05948-f008:**
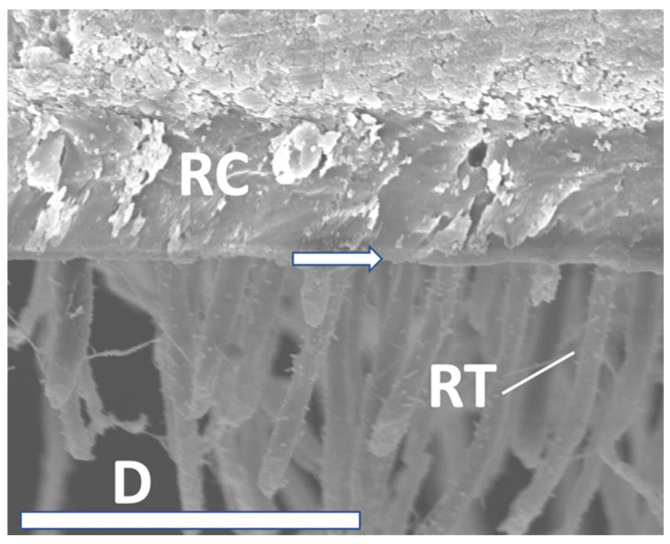
SEM interface (RC: resin composite; RT: resin tags; D: Dentin; hybrid layer: arrow). IBU after 6 months, 2500× magnification/bar: 20 µm.

**Figure 9 materials-16-05948-f009:**
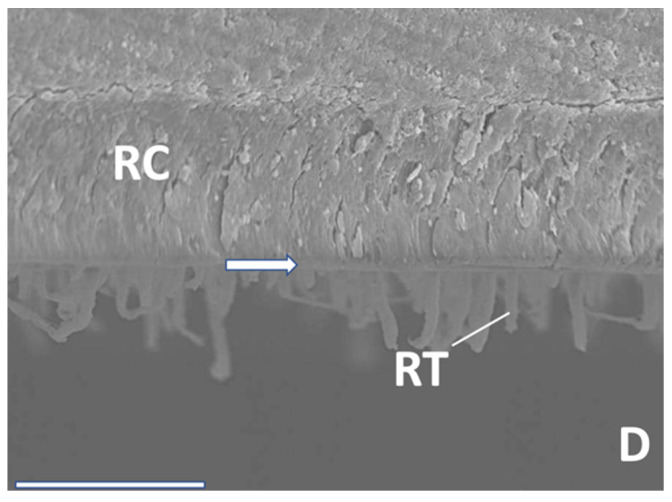
SEM interface (RC: resin composite; RT: resin tags; D: Dentin; hybrid layer: arrow). PBA after 12 months, 1500× magnification/bar: 20 µm.

**Figure 10 materials-16-05948-f010:**
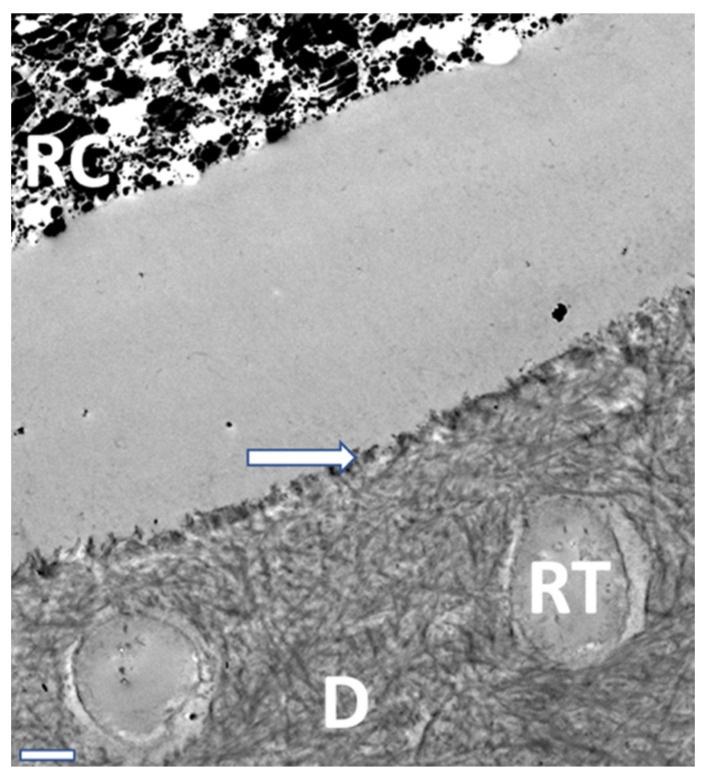
TEM view of the homogenous adhesive layer, the hybrid layer (arrow; approx. 500 nm) and dentinal (D) tubules (filled with resin tags/RT). ABU after 6 months, 7000× magnification/bar: 1 µm.

**Figure 11 materials-16-05948-f011:**
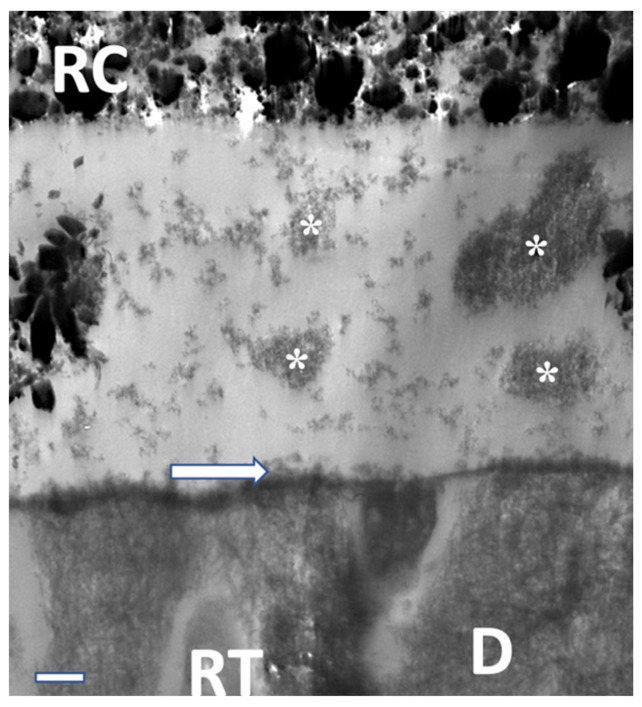
TEM view of the adhesive layer with precipitations (asterisks), hybrid layer (arrow), and dentinal (D) tubules filled with resin tags (RD). AU after 24 h, 7000× magnification/bar: 1 µm.

**Figure 12 materials-16-05948-f012:**
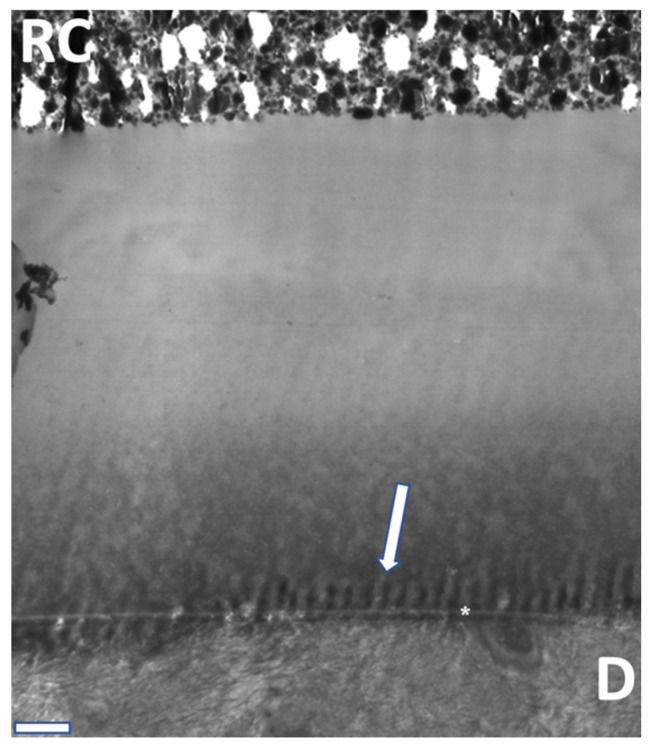
TEM overview of the adhesive layer on dentin (D) with resin composite (RC) with the gradient and the hybrid layer (akterisk). IBU after 12 months. “Piano keys” appearance at the adhesive layer (arrow), 4000× magnification/bar: 1 µm.

**Table 1 materials-16-05948-t001:** Classification of self-etch adhesives and their demineralization capability.

Classification	pH	Demineralization
ultra-mild	>2.5	~300 nm
mild	~2	1 µm
medium	1–2	1–2 µm
strong	<1	>2 µm

**Table 2 materials-16-05948-t002:** Overview of the adhesives, the functional monomer, solvent, and pH value compared with PBN. (a) drying without desiccation, (b) application, (c) application, resting for 10 s, (d) 2 separate applications, (e) rubbing in adhesive for approx. 20 s, (f) air-thinned, (g) 5 s air thinned, (h) 10 s air thinned, (i) 5 s air thinned with maximum air stream, (j) 10 s light cured.

Adhesive	Functional Monomer	Solvent	pH Value	Application
**Prime & Bond^®^ NT (control)** Dentsply Sirona GmbH	UDMA, PENTA	Acetone	2.1	(a), (b), (e), (g), (h)
**iBond^®^ Universal** Kulzer Dental GmbH	10-MDP	Acetone, water	1.6–1.8	(a), (b), (e), (f), (h)
**G-Premio Bond,** GC Europe N.V.	10-MDP	Acetone, 2-Hydroxy-1,3-dimethacryl-oxypropan	2.1	(a), (b), (c), (i), (h)
**Prime & Bond active™** Dentsply Sirona GmbH	10-MDP	Isopropanol, water	2.5	(a), (b), (e), (g), (h)
**Adhese^®^ Universal,** Ivoclar Vivadent	10-MDP	Ethanol, water	2.5–3.0	(a), (b), (e), (h), (j)
**All-Bond Universal^®^,** Bisco Inc.	10-MDP	Ethanol, water	2.5–3.5	(a), (d), (h), (j)

**Table 3 materials-16-05948-t003:** Results of μ-TBS and fracture modes.

** Adhesives/Fracture Mode **	**Control Group** **PBNT**			**IBU**			**GPB**			**PBa**			**AU**			**ABU**		
Storage Period	24 h	6 mo	12 mo	24 h	6 mo	12 mo	24 h	6 mo	12 mo	24 h	6 mo	12 mo	24 h	6 mo	12 mo	24 h	6 mo	12 mo
** µ-TBS (SD) [MPa] **	43.6 (17.7)	32.5 (17.3)	31.2 (20.0)	44.5 (21.2)	31.3 (16.9)	26.9 (18.2)	16.9 (11.3)	15.5 (9.1)	10.9 (10.1)	31.7 (11.0)	30.4 (19.0)	25.0 (22.6)	52.8 (18.3)	44.1 (12.8)	42.9 (19.3)	40.8 (18.7)	41.0 (21.1)	36.3 (16.0)
**Pre-test failures** **[%]**	0.0	0.0	14.6	0.0	0.0	0.0	0.0	4.9	12.2	0.0	3.4	6.2	0.0	0.0	0.0	0.0	4.4	0.0
** Adhesive fractures [%] **	78.8	97.3	85.4	96.9	93.8	97.5	100.0	95.1	87.8	92.5	93.2	86.4	83.1	94.6	86.3	81.3	73.3	88.1
** Cohesive fractures [%] **	21.2	2.7	0.0	3.1	6.2	2.5	0.0	0.0	0.0	7.5	3.4	7.4	16.9	5.4	13.7	18.7	22.2	11.9

## Data Availability

Data from this study are available on reasonable request.

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
