# Peer review of "Dentin Bonding Performance of Universal Adhesives in Primary Teeth In Vitro"

_materials, 2023, doi:10.3390/ma16175948_

Round 1
Reviewer 1 Report
Dear Authors,
I had the chance to review the article "Dentin bonding performance of universal adhesives in primary teeth". Universal adhesives represent the latest generation of bonding systems introduced In the market already some years ago. Their versatility make them suitable in different clinical situations, including the restoration of primary teeth. That is, the objective of the study would be interesting and updated. However, several major revisions have been encountered throughout the text and these should be considered for an overall consideration of the manuscript. According to the many issues found in the text, I'm not able to recommend the article for publication in the present form. I underline some recommendations and suggestions that I hope you can find useful to increase the soundness and reliability of the article.
Best regards
ABSTRACT:
Background: It should be specified that the adhesion test is conducted on which dental substrates of primary dentin?
M&M: Which teeth were used for the study ? Molars or what? This should be specified both here and in the text.
It should be clarified that PBN is used as control.
It is not clear whether the specimens were stored entirely in water during the storage time or they were first cut in sticks and these were then water aged?
The statistical analysis should be also presented in the abstract and significance should be given according to the results obtained.
Results: I would suggest to reorder this part, by indicating per adhesive and storage time statistically significances. In the present way, it seems too confusionary for the reader to easily understand the results. Moreover, there is no indication on IBU results, apart of pretesting failures.
INTRODUCTION
Line 30: are you sure ref 2 is appropriate ?
Lines 31-32: This sentence seems useless to the objective of the study and, moreover, ref 3 seems not to be related to the topic dealt with.
Line 34: please, provide first the entire definition of bonded resin composite and after (BRC).
Lines 34-36: Do you mean evolved in terms of operative simplification, chemical composition and application modes? This should be rightly refined in order to make it more readable.
Line 38. Is the term “the latter” referred to self-etch adhesive? Please, provide a reference supporting this statement.
Lines 38-39: Are you referring to the use of universal adhesive on primary or permanent dentin? I think the references used are not related to this topic, in particular Ref 5. Maybe you can find useful to refer to the following articles: 1) doi: 10.1016/j.dental.2022.01.002 and 2) doi: 10.17796/1053-4625-45.3.7.
Lines 39-42: this part is too schematic and confusionary and should be rearranged. Universal adhesives are intended to be used in different application modes (please, refer to the previously cited ref1) and are different in the chemical composition and acidity that would influence their interaction with the dentin substrates. 1) doi: 10.1111/jerd.12692 and 2) doi: 10.1177/00220345221145673 and 3) doi: 10.1007/s00784-022-04402-3.
Ref 6: Please, check the validity of this reference in supporting this statement.
Lines 43-45: 10-MDP is considered the most reliable functional monomer in gaining effective interaction with dentin and its use is undoubtedly well assessed. However, in my personal opinion, this statement is confusionary, probably with the intent of summarize the efficacy of 10-MDP the Authors are loosening to explain its main benefit. Moreover, references seem to be completely out od theme.
Table 1 is considered useless to the aim of this project. Maybe you can consider to add these information to Table 2
I think the introduction should be more focused on the interaction of universal adhesives on primary dentin, and maybe furnish some information on their application mode. Moreover, the objective of the study should be anticipated by the problems related to universal adhesive on primary dentin in order to justify the aim of the study.
Line 50: You wrote that the test will involve PBN as control…. However, there is no mention in the introduction to understand what PBN is…
MATERIALS AND METHODS
Was the sample size calculated for this study? How was this done?
Line 56: as previously reported, it would be suggested to indicate what kind of primary teeth were used for the study and to what extent the dentin was exposed (middle dentin?).
Table 2: please, I woukld suggest to include the abbreviation into parethesis after the name of the adhesive. Please, provide in the legend the full name of monomers and other abbreviations present in the table. Please, check the format of the table and make it uniformly present over the table.
Were specimen’s preparations performed by a single operator?
Lines 60-62: Did you applied a first layer of composite in 0.5 mm height and then 1 layer of 2 mm for a total eight of 2.5 mm of restoration? Was this reliable for the microtensile test?
Lines 63-66: As mentioned in the abstract, it is not clear to me whether the samples were stored for time 0, 6 and 12 months and then cut at each storage time, or they were first cut into sticks and then the sticks stored? In the first case, could you please provide a reference doing this?
Line 63: was the water substituted over the storage period of time or was maintained always the same?
Lines 66-67: this is a large number of sticks no doubt, also considering the vaste number of groups tested. However, the presentation of the number of sticks tested in general seems useful. Maybe it would be more interesting to indicate, per each group/storage period the number of stick tested.
Fig 1: Please, provide a more exhaustive legend explanation of the steps performed for specimens’ preparation. You presented the number per each step, you maybe find useful to explain each step in the legend, as these were very strictly presented in the text. Consider that the materials and methods section of each study should be well presented in order to make investigators able to reproduce the same test. So all the information should be provided, included type of polymerizing light, wavelength and so on.
Line 64: How were the specimens prepared for fluorescence microscopy? Were all sticks observed for this test or only some representative specimens? Referring to Figs 5-7 seems much more related to rhe results sction. In this section, you should consider including the specimens preparation, not like they resulted after observation.
Do you have a reference of the SEM preparation?
Were the specimens observed under TEM cut and at which dimension?
Figs 11-13 should be adapted to the results section.
Lines 100-101: it is not clear to me the statistical tests used. Please, revise this part.
Table 3 is very confusing and should be reformatted. Moreover, the statistical significance should be indicated per each group. Considering the large number of groups in terms of adhesives and storage period, I would suggest to separate the tables, one per the uTBS results and one for the fracture analysis. The value are mean (standard deviations) not as indicated in the subtitled table. Again, Table 3 should be presented as part of the results section and not in the materials and methods.
Lines 103-105: Did you also have mixed failures? Mixed failures are the most encountered type of fractures occurring in dental adhesion. Howevr, it seems you only observed the A and C ones? And the C were present inside the adhesiove or composite or dentin?
Fig. 2 is very confusionary. I cannot understand the different level of significance as this was not well informative in the materials and methods section.
Results section is very confusionary. I would suggest to reorder. The number of groups per each testing period is really huge and this make the work not easy. But for the reader, these information should be provided in a more simple way, in order to easily understand the results of your test.
Line 160. “Fluorescence microscopy proved to be a reliable tool for exact determination… “ seems too generous… the results section should only provide the exact presentation of your results trying to avoid any kind of consideration that can, instead, be made in the discussion section.
Line 177: Again, bond strength to what type of substrate?
Discussion is poor and not well focused on the results obtained in the test.
References need a throughout check.
In general, I would suggest a widespread English revision throughout the text and double-check editing of typos.
Author Response
Reviewer 1 (Rebuttal after >>>):
Background: It should be specified that the adhesion test is conducted on which dental substrates of primary dentin?
>>> information added. Thank you.
Which teeth were used for the study ? Molars or what? This should be specified both here and in the text.
>>> information added. Thank you.
It should be clarified that PBN is used as control.
>>> information added. Thank you.
It is not clear whether the specimens were stored entirely in water during the storage time or they were first cut in sticks and these were then water aged?
>>> information added. Thank you.
The statistical analysis should be also presented in the abstract and significance should be given according to the results obtained.
>>> information added. Thank you.
I would suggest to reorder this part, by indicating per adhesive and storage time statistically significances. In the present way, it seems too confusionary for the reader to easily understand the results. Moreover, there is no indication on IBU results, apart of pretesting failures.
>>> Figures and tables were completely rearranged, incomplete groups were deleted. Thank you.
Line 30: are you sure ref 2 is appropriate ?
>>> No, it was not, shame on the PhD student, my fault overlooking it. Freeware reference program full of errors. So, the cited literature was completely rearranged. Thank you.
Lines 31-32: This sentence seems useless to the objective of the study and, moreover, ref 3 seems not to be related to the topic dealt with.
>>> Literature was completely rearranged.
Line 34: please, provide first the entire definition of bonded resin composite and after (BRC).
>>> information added. Thank you.
Lines 34-36: Do you mean evolved in terms of operative simplification, chemical composition and application modes? This should be rightly refined in order to make it more readable.
>>> information added. Thank you.
Line 38. Is the term “the latter” referred to self-etch adhesive? Please, provide a reference supporting this statement.
>>> information added. Thank you.
Lines 38-39: Are you referring to the use of universal adhesive on primary or permanent dentin? I think the references used are not related to this topic, in particular Ref 5. Maybe you can find useful to refer to the following articles: 1) doi: 10.1016/j.dental.2022.01.002 and 2) doi: 10.17796/1053-4625-45.3.7.
>>> references were completely rearranged and changed where necessary.
Lines 39-42: this part is too schematic and confusionary and should be rearranged. Universal adhesives are intended to be used in different application modes (please, refer to the previously cited ref1) and are different in the chemical composition and acidity that would influence their interaction with the dentin substrates. 1) doi: 10.1111/jerd.12692 and 2) doi: 10.1177/00220345221145673 and 3) doi: 10.1007/s00784-022-04402-3.
>>> This part was rewritten. Thank you.
Ref 6: Please, check the validity of this reference in supporting this statement.
>>> checked. Was wrong. Thank you.
Lines 43-45: 10-MDP is considered the most reliable functional monomer in gaining effective interaction with dentin and its use is undoubtedly well assessed. However, in my personal opinion, this statement is confusionary, probably with the intent of summarize the efficacy of 10-MDP the Authors are loosening to explain its main benefit. Moreover, references seem to be completely out of theme.
>>> information added, references changed. Thank you.
Table 1 is considered useless to the aim of this project. Maybe you can consider to add these information to Table 2
>>> we see this differently. Table 2 was addressed by a few reviewers of the seven reviewers that I have to answer, so we had to decide which of the seven reviewers we follow.
I think the introduction should be more focused on the interaction of universal adhesives on primary dentin, and maybe furnish some information on their application mode. Moreover, the objective of the study should be anticipated by the problems related to universal adhesive on primary dentin in order to justify the aim of the study.
>>> The introduction was completely rewritten and literatur work was thoroughly revised as well. Thank you.
Line 50: You wrote that the test will involve PBN as control…. However, there is no mention in the introduction to understand what PBN is…
>>> information added. Thank you.
Was the sample size calculated for this study? How was this done?
>>> information added. Thank you.
Line 56: as previously reported, it would be suggested to indicate what kind of primary teeth were used for the study and to what extent the dentin was exposed (middle dentin?).
>>> information added. Thank you.
Table 2: please, I woukld suggest to include the abbreviation into parethesis after the name of the adhesive. Please, provide in the legend the full name of monomers and other abbreviations present in the table. Please, check the format of the table and make it uniformly present over the table.
>>> information added. Thank you.
Were specimen’s preparations performed by a single operator?
>>> information added. Thank you.
Lines 60-62: Did you applied a first layer of composite in 0.5 mm height and then 1 layer of 2 mm for a total eight of 2.5 mm of restoration? Was this reliable for the microtensile test?
>>> information added. Thank you.
Lines 63-66: As mentioned in the abstract, it is not clear to me whether the samples were stored for time 0, 6 and 12 months and then cut at each storage time, or they were first cut into sticks and then the sticks stored? In the first case, could you please provide a reference doing this?
>>> information added. Thank you.
Line 63: was the water substituted over the storage period of time or was maintained always the same?
>>> information added. Thank you, Sherlock – excellent point!
Lines 66-67: this is a large number of sticks no doubt, also considering the vaste number of groups tested. However, the presentation of the number of sticks tested in general seems useful. Maybe it would be more interesting to indicate, per each group/storage period the number of stick tested.
>>> information added. Thank you.
Fig 1: Please, provide a more exhaustive legend explanation of the steps performed for specimens’ preparation. You presented the number per each step, you maybe find useful to explain each step in the legend, as these were very strictly presented in the text. Consider that the materials and methods section of each study should be well presented in order to make investigators able to reproduce the same test. So all the information should be provided, included type of polymerizing light, wavelength and so on.
>>> information added. Thank you.
Line 64: How were the specimens prepared for fluorescence microscopy? Were all sticks observed for this test or only some representative specimens? Referring to Figs 5-7 seems much more related to rhe results sction. In this section, you should consider including the specimens preparation, not like they resulted after observation.
>>> The specimens do not have to be prepared for FM.
Do you have a reference of the SEM preparation?
>>> information added. Thank you.
Were the specimens observed under TEM cut and at which dimension?
>>> information added. Thank you.
Figs 11-13 should be adapted to the results section.
>>> information added. Thank you.
Lines 100-101: it is not clear to me the statistical tests used. Please, revise this part.
>>> information added. Thank you.
Table 3 is very confusing and should be reformatted. Moreover, the statistical significance should be indicated per each group. Considering the large number of groups in terms of adhesives and storage period, I would suggest to separate the tables, one per the uTBS results and one for the fracture analysis. The value are mean (standard deviations) not as indicated in the subtitled table. Again, Table 3 should be presented as part of the results section and not in the materials and methods.
>>> Tables and figures were completely changed and information density reduced. Thank you.
Lines 103-105: Did you also have mixed failures? Mixed failures are the most encountered type of fractures occurring in dental adhesion. Howevr, it seems you only observed the A and C ones? And the C were present inside the adhesiove or composite or dentin?
>>> Mixed failures were the missing % in the table. Due to their minority, they were not separately mentioned.
Fig. 2 is very confusionary. I cannot understand the different level of significance as this was not well informative in the materials and methods section.
>>> Figure 2 was completely changed.
Results section is very confusionary. I would suggest to reorder. The number of groups per each testing period is really huge and this make the work not easy. But for the reader, these information should be provided in a more simple way, in order to easily understand the results of your test.
>>> reordered. Thank you.
Line 160. “Fluorescence microscopy proved to be a reliable tool for exact determination… “ seems too generous… the results section should only provide the exact presentation of your results trying to avoid any kind of consideration that can, instead, be made in the discussion section.
>>> information added, passage clarified. Thank you.
Line 177: Again, bond strength to what type of substrate?
>>> information added. Thank you.
Discussion is poor and not well focused on the results obtained in the test.
>>> Also discussion was completely rewritten.
References need a throughout check.
>>> done.
Reviewer 2 Report
The manuscript "Dentin bonding performance of universal adhesives in primary teeth" is a relevant study on the performance of varying universal adhesives applied using the self-etch mode. The findings of this study are interesting, showing importance to clinicians dealing with adhesive restorations in primary teeth. The study was well conducted, following a rigorous methodology. The authors used appropriate tests to evaluate their research goals. Congratulations to the authors. The manuscript is worth of publication in Materials.
Author Response
Reviewer 2
The manuscript "Dentin bonding performance of universal adhesives in primary teeth" is a relevant study on the performance of varying universal adhesives applied using the self-etch mode. The findings of this study are interesting, showing importance to clinicians dealing with adhesive restorations in primary teeth. The study was well conducted, following a rigorous methodology. The authors used appropriate tests to evaluate their research goals. Congratulations to the authors. The manuscript is worth of publication in Materials.
Rebuttal: Thank you very much.
Reviewer 3 Report
Interesting study about the effect of water storage on the bond strength of different adhesives.
Some points could be improved before publication:
- Line 42: This table would be better being at the Discussion instead of Introduction section.
- Line 66: What was the speed of the micro-tensile testing?
- Table 2: the "Application" column could be improved. It is hard to keep going back to the Table description to understand the application steps of each adhesive.
- Table 3 and Figure 2: please, correct the word "kohesive".
- Table 3 and Figure 2: there is no explanation for the missing values for 6m CUB, 6m and 12m CUBQ.
- Pages 7-end: track-changes active showing text changes. This should be a "clean version". Hard to review properly.
- Figure "413" not necessary. (Figure of the 10-MDP molecular configuration)
- It would be better having a wider discussion comparing the results of different adhesives techniques used in this study.
Minor errors.
Author Response
Reviewer 3 (Rebuttal after >>>)
Interesting study about the effect of water storage on the bond strength of different adhesives.
>>> Thank you.
- Line 42: This table would be better being at the Discussion instead of Introduction section.
>>> 5 our of seven reviewers criticized that the tables are too confusing, leaving Table1 there helps and represents the majority of the 7 reviewers.
- Line 66: What was the speed of the micro-tensile testing?
>>> information added. Thank you.
- Table 2: the "Application" column could be improved. It is hard to keep going back to the Table description to understand the application steps of each adhesive.
>>> This is a standard procedure. Not changed.
- Table 3 and Figure 2: please, correct the word "kohesive".
>>> Changed. Thank you.
- Table 3 and Figure 2: there is no explanation for the missing values for 6m CUB, 6m and 12m CUBQ.
>>> These groups were completely dropped. It as an accident in the lab but we cannot explain this in a scientific way.
- Pages 7-end: track-changes active showing text changes. This should be a "clean version". Hard to review properly.
>>> This is surely addressed to mdpi.
- Figure "413" not necessary. (Figure of the 10-MDP molecular configuration)
>>> figure deleted. Thank you.
- It would be better having a wider discussion comparing the results of different adhesives techniques used in this study.
>>> Both introduction and discussion were completely rewritten. Thank you.
Reviewer 4 Report
This is an interesting manuscript reporting an in-vitro study aimed to evaluate micro-tensile bond strength of universal adhesives in primary dentition after different storage periods. To improve the manuscript, here are my suggestions/modifications listed below:
- Study design should be indicated in the title: “in vitro study”.
- Number of adhesive bonding systems used should be indicated in the title.
- As title indicated, the aim of study should specify “primary teeth” in the abstract.
- More details on water storage of specimens should be added to the abstract
- Please cite appropriate reference for the provided “Classification of self-etch adhesives and their demineralization capability” in table 1.
- Kindly add examples for each class to table 1 (additional column needed).
- Extracted primary teeth were stored in chloramine-T for “less than 4 weeks”. This should be more specific.
- More details on the characteristics of the teeth than just “120 freshly extracted primary teeth” should be added.
- Inclusion and exclusion criteria of the primary teeth included in the experiment should be described.
- More details on preparation of samples and bonding stages should be added to the methods section.
- Standardization of all samples should be described.
- Details on μ-TBS test should be added.
- Figure 1 legend lacks detailed description regarding each stage of the procedure.
- Kindly, add a footer including the definition for all abbreviations used in table 3
- If possible, please replace figures 4, 5, and 6 with better-quality figures.
- Please add a straight-to-point take-home message to the conclusions section.
Author Response
Reviewer 4 (rebuttal after >>>)
This is an interesting manuscript reporting an in-vitro study aimed to evaluate micro-tensile bond strength of universal adhesives in primary dentition after different storage periods. To improve the manuscript, here are my suggestions/modifications listed below:
>>> Thank you.
- Study design should be indicated in the title: “in vitro study”.
>>> information added. Thank you.
- Number of adhesive bonding systems used should be indicated in the title.
>>> contradictory recommendations of different reviewers. We did not change that.
- As title indicated, the aim of study should specify “primary teeth” in the abstract.
>>> information added. Thank you.
- More details on water storage of specimens should be added to the abstract
>>> information added. Thank you.
- Please cite appropriate reference for the provided “Classification of self-etch adhesives and their demineralization capability” in table 1.
>>> Done. Thank you.
- Kindly add examples for each class to table 1 (additional column needed).
>>> We left this information in the discussion.
- Extracted primary teeth were stored in chloramine-T for “less than 4 weeks”. This should be more specific.
>>> information not added because it is not possible.
- More details on the characteristics of the teeth than just “120 freshly extracted primary teeth” should be added.
>>> information added. Thank you.
- Inclusion and exclusion criteria of the primary teeth included in the experiment should be described.
>>> information added. Thank you.
- More details on preparation of samples and bonding stages should be added to the methods section.
>>> information added. Thank you.
- Standardization of all samples should be described.
>>> information added. Thank you.
- Details on μ-TBS test should be added.
>>> information added. Thank you.
- Figure 1 legend lacks detailed description regarding each stage of the procedure.
>>> information added. Thank you.
- Kindly, add a footer including the definition for all abbreviations used in table 3
>>> information not added, too many repetitions otherwise.
- If possible, please replace figures 4, 5, and 6 with better-quality figures.
>>> Why? The quality is very good.
- Please add a straight-to-point take-home message to the conclusions section.
>>> information added. Thank you.
Reviewer 5 Report
The reviewer really appreciates the efforts of the authors to conduct this study. The study design is good enough to extract valuable conclusions. However, there are some scopes to improve the quality of the manuscript. The reviewer would like to suggest the following revision in the manuscript to make it suitable for publication.
· The introduction part is too short and missing logical background of the current research
· The authors can add a review of previously published literature in the introduction section to explain the rationale of the present study
· Why there are missing groups in CUB, and CUBQ?
· The author used different abbreviations for the groups in the abstract and in the result.
· Some abbreviation is missing
· Statistics need revision. There are two factors in this study (materials and time). The author should use Two-Way ANOVA to see the interaction. One-Way ANOVA and T-test (group CUB) for comparison within the groups at different time periods. Multiple comparisons of 24h data among the tested groups
· Please correct the spelling of Kohesive to cohesive failure.
· Table 3 is too congested. It can be removed as the same information repeated in figure 2 and 3
· Please adjust the dimension of Figure 2 to make it readable
· Please make composite images for failure mode, SEM, and TEM for a better understanding instate of the individual figure.
· The author put null hypothesis after objectives however, in the discussion did not mention whether the null hypothesis was accepted or rejected
· The discussion part is short and an explanation of obtained result is necessary.
The conclusion should mention the significant outcomes of the study. The author can add a few more lines
minor revision required
Author Response
Reviewer 5 (Rebuttal after >>>)
The reviewer really appreciates the efforts of the authors to conduct this study. The study design is good enough to extract valuable conclusions. However, there are some scopes to improve the quality of the manuscript. The reviewer would like to suggest the following revision in the manuscript to make it suitable for publication.
>>> Thank you.
- The introduction part is too short and missing logical background of the current research
>>> The introduction was completely rewritten, literature was changed.
- The authors can add a review of previously published literature in the introduction section to explain the rationale of the present study
>>> information added. Thank you.
- Why there are missing groups in CUB, and CUBQ?
>>> These groups got lost in a lab accident, therefore we decided to skip them.
- The author used different abbreviations for the groups in the abstract and in the result.
>>> information added. Thank you.
- Some abbreviation is missing
>>> information added. Thank you.
- Statistics need revision. There are two factors in this study (materials and time). The author should use Two-Way ANOVA to see the interaction. One-Way ANOVA and T-test (group CUB) for comparison within the groups at different time periods. Multiple comparisons of 24h data among the tested groups
>>> Out statistician did not agree here.
- Please correct the spelling of Kohesive to cohesive failure.
>>> information added. Thank you.
- Table 3 is too congested. It can be removed as the same information repeated in figure 2 and 3
>>> Changed. Thank you.
- Please adjust the dimension of Figure 2 to make it readable
>>> Done accordingly. Thank you.
- Please make composite images for failure mode, SEM, and TEM for a better understanding instate of the individual figure.
>>> Good point but it would be barely readable then. Not changed.
- The author put null hypothesis after objectives however, in the discussion did not mention whether the null hypothesis was accepted or rejected
>>> information added. Thank you.
- The discussion part is short and an explanation of obtained result is necessary.
>>> The discussion was rewritten. Thank you.
The conclusion should mention the significant outcomes of the study. The author can add a few more lines
>>> information added. Thank you.
Reviewer 6 Report
Abstract
Bacground:Need to add the existiong problem but aim is there
Discussion
Why again aim of the study is added
Abstract
Bacground:Need to add the existiong problem but aim is there
Discussion
Why again aim of the study is added
Author Response
Reviewer 6 (Rebuttal after >>>)
Background:Need to add the existiong problem but aim is there
>>> information added. Thank you.
Why again aim of the study is added
>>> information added. Thank you.
Comments on the Quality of English Language
>>> The paper was completely rewritten. Thank you.
Background: Need to add the existiong problem but aim is there
>>> information added. Thank you.
Why again aim of the study is added
>>> Because it is common use. Thank you.
Reviewer 7 Report
General:
1. A native editor should revise the whole text. It is pretty challenging to understand the text.
Title:
1. The study type must be included in the title.
2. The aim of the study is not matched with the title.
Abstract:
1. The methodology within the abstract should be expanded. It does not provide sufficient details.
Keywords:
1. It is recommended to write keywords in an alphabetic order.
Introduction:
1. Regardless of the abbreviations mentioned in the abstract, those abbreviations in the text should be defined at their first appearance. Please revise the text based on this comment.
2. The introduction is relatively poor. Authors are encouraged to expand this section, focusing on the importance of performing successfully bonded restorations for primary teeth, the importance of bonding agents, different properties of bonding agents that affect the bonding strength, etc.
3. The aim of this study is not well-described in the introduction. The authors presented no data regarding the importance of storage in bonding agents. I advise re-writing the whole introduction in a well-organized manner focusing on bonding agents and their role in restoring primary teeth.
Materials and Methods:
1. What kind of primary teeth have been used in this study? Primary molars and incisors could present different characteristics regarding bond strength. Were all teeth similar?
2. The materials' brands are not placed right after them. Please revise them.
3. This section is quite challenging to follow. I suggest re-writing the methods used using additional details.
4. What was the brand of the composite material used?
5. Authors should describe why different application types have been used for each bonding agent and how this could affect the results.
6. Figure 1 should be accompanied by texts describing each step.
Results:
1. TOOOOOOOOOO many grammatical and typo errors. It was pretty unreadable.
Discussion:
1. The explanation regarding the bonding agents is quite acceptable. However, the authors should provide a real discussion comparing the results of this study with the previous ones and discussing their differences and different parameters that may have affected the results, etc.
2. Limitations of the study should be added.
First, the authors should re-write the manuscript in a more scientific way. Then, a native editor should revise the text.
Author Response
Reviewer 7 (Rebuttal after >>>):
General:
A native editor should revise the whole text. It is pretty challenging to understand the text.
>>> The text was completely rewritten.
The study type must be included in the title.
>>> done.
The aim of the study is not matched with the title.
>>> changed.
The methodology within the abstract should be expanded. It does not provide sufficient details.
>>> changed.
It is recommended to write keywords in an alphabetic order.
>>> I never did this, but thank you.
Regardless of the abbreviations mentioned in the abstract, those abbreviations in the text should be defined at their first appearance. Please revise the text based on this comment.
>>> done.
The introduction is relatively poor. Authors are encouraged to expand this section, focusing on the importance of performing successfully bonded restorations for primary teeth, the importance of bonding agents, different properties of bonding agents that affect the bonding strength, etc.
>>> the introduction was completely rewritten.
The aim of this study is not well-described in the introduction. The authors presented no data regarding the importance of storage in bonding agents. I advise re-writing the whole introduction in a well-organized manner focusing on bonding agents and their role in restoring primary teeth.
>>> the introduction was completely rewritten.
What kind of primary teeth have been used in this study? Primary molars and incisors could present different characteristics regarding bond strength. Were all teeth similar?
>>> information added. Thank you.
The materials' brands are not placed right after them. Please revise them.
>>> done.
This section is quite challenging to follow. I suggest re-writing the methods used using additional details.
>>> Also the M&M section was completely rewritten with several details having been added.
What was the brand of the composite material used?
>>> information added
Authors should describe why different application types have been used for each bonding agent and how this could affect the results.
>>> information added.
Figure 1 should be accompanied by texts describing each step.
>>> some information added without confusing the reader.
TOOOOOOOOOO many grammatical and typo errors. It was pretty unreadable.
>>> The whole mamuscript was rearranged and rewritten.
The explanation regarding the bonding agents is quite acceptable. However, the authors should provide a real discussion comparing the results of this study with the previous ones and discussing their differences and different parameters that may have affected the results, etc.
>>> Discussion was rearranged and rewritten to meet the reviewers aspects.
Limitations of the study should be added.
>>> done.
Round 2
Reviewer 5 Report
Thank you for the revision
Author Response
The reviewer wrote that both introduction and discussion should be optimized. This was done.
Reviewer 7 Report
I am still not convinced by the description of the methodology used for bonding specimens. Authors are encouraged to describe this section quite clearly. How were the specimens bonded? Their rubbing method, light curing, etc.
How was the sample size estimated?
Figure 1: still not satisfactory. Authors should describe each step like this: (1): XXX, (2): YYY, (3): ZZZ, …..
This sentence is needed to be clearly described: After debonding, fractures were analyzed under a fluorescence microscope at 40x magnification.
The quality of the text still is not satisfactory. As I have suggested, a native editor should revise the whole text.
A native editing is a must for this manuscript.
Author Response
Remark: I am still not convinced by the description of the methodology used for bonding specimens. Authors are encouraged to describe this section quite clearly. How were the specimens bonded? Their rubbing method, light curing, etc.
Answer: This is now referenced to Table 2.
Remark: How was the sample size estimated?
Answer: As in referenced previous studies, we took the recommendations of the Academy of Dental Materials for micro tensile testing.
Remark: Figure 1: still not satisfactory. Authors should describe each step like this: (1): XXX, (2): YYY, (3): ZZZ, …..
Answer: Completely changed.
Remark: This sentence is needed to be clearly described: After debonding, fractures were analyzed under a fluorescence microscope at 40x magnification.
Answer: Information added.
Remark: The quality of the text still is not satisfactory. As I have suggested, a native editor should revise the whole text.
Answer: Although reviewer #1 estimated the English as perfect, the paper was again completely corrected by an English native.